# Transarterial Chemoembolization (TACE) Plus Sorafenib Compared to TACE Alone in Transplant Recipients with Hepatocellular Carcinoma: An Institution Experience

**DOI:** 10.3390/cancers14030650

**Published:** 2022-01-27

**Authors:** Maen Abdelrahim, David Victor, Abdullah Esmail, Sudha Kodali, Edward A. Graviss, Duc T. Nguyen, Linda W. Moore, Ashish Saharia, Robert McMillan, Joy N. Fong, Ahmed Uosef, Mahmoud Elshawwaf, Kirk Heyne, Rafik M. Ghobrial

**Affiliations:** 1Section of GI Oncology, Department of Medical Oncology, Houston Methodist Cancer Center, Houston, TX 77030, USA; AEsmail@houstonmethodist.org (A.E.); Heyne@houstonmethodist.org (K.H.); 2Cockrell Center of Advanced Therapeutics Phase I Program, Houston Methodist Research Institute, Houston, TX 77030, USA; 3Cancer Clinical Trials, Houston Methodist Research Institute, Houston, TX 77030, USA; eagraviss@houstonmethodist.org (E.A.G.); dtnguyen6@houstonmethodist.org (D.T.N.); Auosef@houstonmethodist.org (A.U.); MElshawwaf@houstonmethodist.org (M.E.); 4Weill Cornell Medical College, New York, NY 10021, USA; ASaharia@houstonmethodist.org (A.S.); rrmcmillan@houstonmethodist.org (R.M.); RMGhobrial@houstonmethodist.org (R.M.G.); 5J.C. Walter Jr. Center for Transplantation, Sherrie and Alan Conover Center for Liver Disease and Transplantation, Houston, TX 77030, USA; dwvictor@houstonmethodist.org (D.V.); SKodali@houstonmethodist.org (S.K.); lwmoore@houstonmethodist.org (L.W.M.); jvnoltefong@houstonmethodist.org (J.N.F.); 6Department of Pathology and Genomic Medicine, Houston Methodist Hospital, Houston, TX 77030, USA

**Keywords:** hepatocellular carcinoma, TACE, immunotherapy, LT, TKI, Milan criteria, sorafenib

## Abstract

**Simple Summary:**

Hepatocellular carcinoma (HCC) is the sixth most common malignancy and the third most common cause of cancer-related mortality worldwide. Transarterial chemoembolization has shown survival benefits in patients with early to intermediate-stage HCC, becoming the standard of care and recommended treatment modality by most clinical practice guidelines. The purpose of this current study was to compare the outcomes of HCC transplant candidates treated at our institution with TACE combined with sorafenib versus TACE monotherapy, which will provide further evidence for clinical practice. This study found that using TACE plus sorafenib is generally well-tolerated and associated with improved overall survival in transplant recipient patients with unresectable HCC. A multi-center and prospective study is needed. Randomized and controlled trials are needed to confirm these preliminary findings.

**Abstract:**

Background: Hepatocellular carcinoma (HCC) is the sixth most common malignancy and the third most common cause of cancer-related mortality worldwide. Transarterial chemoembolization has shown survival benefits in patients with early to intermediate-stage HCC, becoming the standard of care and recommended treatment modality by most clinical practice guidelines. The most recent trials of the TACE plus sorafenib combined therapy in patients with unresectable HCC have yielded inconsistent outcomes. The purpose of this study was to compare the outcomes of HCC patients treated with the TACE sorafenib combination as opposed to TACE monotherapy. Methods: This retrospective study included all patients with unresectable HCC who underwent liver transplantation and were treated by either TACE alone or TACE plus sorafenib between July 2008–December 2019. Demographic and clinical data as well as HCC recurrence post-liver transplant (LT) were reported as frequencies and proportions for categorical variables and as the median and interquartile range (IQR) or mean. Chi-square or Fisher’s exact tests were performed for categorical variables and the Kruskal-Wallis test or unpaired test was performed for continuous variables. Kaplan-Meier curves present overall patient survival and HCC-free survival. Results: A total of 128 patients received LT, with a median (IQR) age of 61.4 (57.0, 66.3) years; most were males (77%). Within the TACE-only group, 79 (77%) patients met the Milan criteria and 24 (23%) were beyond the Milan criteria, while the TACE plus sorafenib group had a higher proportion of patients beyond the Milan criteria: 16 (64%) vs. 9 (36%); *p* = 0.01. The five-year disease-free survival (DFS) between the treatment groups approached significance, with 100% DFS in the TACE plus sorafenib group vs. 67.2% in the TACE-alone group, *p* = 0.07. Five-year patient survival was 77.8% in the TACE plus sorafenib group compared to 61.5% in the TACE-alone group (*p* = 0.51). However, in patients who met the beyond Milan criteria, those who received TACE alone had a lower average amount of (percent) tumor necrosis on explant pathology (43.8% ± 32%) compared to patients who received TACE plus sorafenib (69.6% ± 32.8%, *p* = 0.03). Conclusion: This study identified that using TACE plus sorafenib is generally well-tolerated and demonstrated improved overall survival compared to TACE only in transplant recipients with unresectable HCC. A multi-center and prospective randomized controlled trial is needed to substantiate these findings.

## 1. Introduction

Hepatocellular carcinoma (HCC) is the sixth most common malignancy worldwide and the third cause of cancer-related mortality [1]. It is the second most frequent liver malignancy following liver metastasis and the most frequent primary liver neoplasm, accounting for more than 850,000 new cases each year and more than 800,000 deaths [2]. According to the National Cancer Institute’s SEER database, the average five-year survival rate of HCC patients in the US is 19.6%; however, it can be lower, reaching 2.5% for advanced, metastatic disease [3]. Curative treatments for early HCC include surgical resection, liver transplantation, or local ablation therapy [4]. Unfortunately, most patients are diagnosed at an advanced stage and are not candidates for curative treatment options [5].

Transarterial chemoembolization (TACE) has shown survival benefits in patients with early to intermediate HCC stages. TACE has become the standard of care and the recommended treatment modality by most clinical practice guidelines, including the Barcelona Clinic Liver Cancer (BCLC) staging classification [6], the European Association for the Study of the Liver [7], the American Association for the Study of Liver Diseases (AASLD) [8,9], as well as other hepatology societies [10,11]. Due to large tumor volumes and high progression rates, TACE procedures often occur multiple times in a single patient. Some data reported that multiple TACE treatments may lead to the deterioration of liver function, which is usually associated with worse outcomes and poor patient prognoses [12]. Moreover, the efficacy of TACE declines significantly with supplementary TACE procedures, with progressive disease rates of 18%, 21%, 25%, and 27% for the first, second, third, and fourth TACE procedures, respectively [13]. This “TACE refractory” concept has been proposed and discussed in the recent literature [14,15].

The treatment of “TACE-refractory” HCC has attracted the attention of gastrointestinal oncologists as well as interventional radiologists. Clinical studies aimed at improving TACE efficacy have sought to combine TACE with other liver-directed and systemic therapies, including ablation (such as radiofrequency or microwave ablation), radiotherapy (such as stereotactic radiotherapy or radioactive particle seeding), multi-kinase inhibitors (such as sorafenib), and immunotherapy [16,17,18].

In addition, it has been shown that TACE increases tumor hypoxia and the expression of hypoxia-inducible factor 1α (*HIF-1α*) [19,20,21]. Increased HIF-1α results in increased tumor angiogenesis mediated by the up-regulation of pro-angiogenic factors, including vascular endothelial growth factor (*VEGF*) and platelet-derived growth factor (*PDGF*) [19,20,21]. It is commonly believed that blockading the *VEGF* pathway may prevent the effects of a TACE-induced surge in pro-angiogenic factors [19,20,21,22]. Preclinical data have shown that the combination of antiangiogenic therapy with TACE decreases vessel density and tumor volume, which leads to increased survival when compared with TACE alone [22].

Sorafenib is an oral agent that is classified as a multi-kinase inhibitor that inhibits tumor angiogenesis by targeting the *RAF**-MEK**-ERK* signaling pathway or by blocking the expression of vascular endothelial growth factor receptor (*VEGFR*). Thus, Sorafenib possesses antiangiogenic and direct anti-tumor effects. For this reason, sorafenib may suppress the surge of proangiogenic factors occurring after TACE. Trials assessing the combination of TACE plus sorafenib in patients with unresectable HCC have yielded inconsistent results. A recent study has shown that adding sorafenib to TACE significantly prolongs the recurrence of HCC [23]. In contrast, other trials (SPACE and TACE2) [24,25] did not show significant benefit initially; however, later subgroup analyses suggested that a longer duration of treatment with sorafenib in combination with TACE may improve clinical outcomes [26]. A meta-analysis of randomized trials conducted by Facciorusso et al. [27] showed that after chemoembolization there was a statistically significant increase in severe toxicity (risk ratio: 1.44, 1.08–1.92, *p* = 0.01), though this outcome might be influenced by the heterogeneity of the procedures used.

The liver is the only solid organ that has an extra treatment option (i.e., liver transplantation (LT) for HCC) along with established options such as surgical therapies in cases who reported with well-compensated liver function in addition to radiofrequency ablation in those diagnosed with small tumors [28]. However, in 90% of patients, HCC is reported in the setting of liver cirrhosis diseases [28], in which case optimal management is LT with a five-year survival rate of approximately 80% [28]. Therefore, LT offers an optimal treatment option in a specific subgroup of patients with HCC [28]. The eligibility criteria can consider many factors besides the tumor size and numbers, such as tumor biology (including alpha-fetoprotein (AFP) concentration), but tumor size is usually the obstacle for HCC patients along with the availability of donor organs and the composition of the waitlist [28]. The combination of TACE and sorafenib can help to downstage patients with unresectable HCC who do not meet the tumor size requirement for LT. In addition, TACE plus TKI provides tumor stability for HCC patients listed for LT. These stepwise, together with some improvements and modifications in the eligibility criteria at some transplant centers, would help the response to the increasing incidence of HCC and provide improved survival results.

Therefore, whether the combination of TACE and sorafenib in patients with unresectable HCC awaiting LT can improve survival outcomes remains unclear, and further studies are needed. The purpose of this current study was to compare the outcomes of HCC transplant candidates treated at our institution with TACE combined with sorafenib versus TACE monotherapy, which will provide further evidence for clinical practice.

## 2. Materials and Methods

### 2.1. Participants

All patients who received a liver transplant between July 2008–December 2019 underwent screening for eligibility. Patients who had HCC as either a primary or secondary diagnosis reported to the United Network for Organ Sharing. The diagnosis of HCC consisted primarily of a radiographic review and evidence of pathological findings that confirmed the diagnosis. Patients were divided into two treatment groups: (1) those with TACE alone and (2) those who received TACE plus TKI. Patients were then stratified based on tumor size criteria of meeting either the Milan or beyond Milan criteria.

Exclusion criteria included multi-organ, re-transplantations, patients who did not undergo at least one TACE procedure prior to the transplant, and patients who received other forms of locoregional therapy, including microwave ablation, radiofrequency ablation, radioembolization, or yttrium-90. Patients who met the eligibility criteria were then stratified based on the receipt of at least one cycle of TKI agents, such as sorafenib (Nexavar) (Bayer Healthcare Pharmaceuticals Inc., Whippany, NJ, USA) prior to LT.

### 2.2. Neoadjuvant/Adjuvant Therapy

All patients followed routine protocols for neoadjuvant and adjuvant therapy. TACE procedures were performed by interventional radiologists. Specialized liver radiologists examined computed tomography (CT) and magnetic resonance imaging (MRI) scans for tumor size and characteristics. A multi-disciplinary team of GI oncologists, transplant surgeons, radiologists, and hepatologists reviewed each case to determine locoregional therapy (LRT) and systemic therapy regimens. Tumor burden and disease multifocality were taken into consideration, as were restaging scans and responses to prior therapies.

### 2.3. TACE and TKI

Since TACE procedures are considered minimally invasive treatments, procedures were usually performed in an interventional radiology suite or, in rare cases, in the operating room by our highly trained interventional radiologists. The course of blood arteries feeding the tumor was mapped using X-ray imaging. The route was also mapped using contrast material. In some cases, allopurinol was prescribed to protect the kidneys from chemotherapy and the metabolites created by dead tumor cells. In addition, some medications were prescribed to help prevent nausea and pain. Prophylactic antibiotics were also given to prevent infection.

A sedative agent was administered through an intravenous (IV) line into a vein in the arm or hand. Some patients required general anesthesia. A small incision at the surgical site was opened by the interventional radiologist. The interventional radiologist guided a small catheter through the opening into an artery in the groin and progressed it to the liver using X-ray guidance. The contrast substance was then administered through the catheter and another set of X-rays were taken.

The Seldinger method was used to puncture the femoral artery. Using a 5 Fr catheter (Elway, Terumo, Tokyo, Japan), angiography of the celiac, hepatic, and superior mesenteric arteries was conducted to detect all tumor-feeding arteries. Following the identification of the target artery, the tumor-feeding arteries were catheterized using a 2.3 Fr to 2.8 Fr tip microcatheter, Progreat^TM^ Terumo or Renegade^TM^ HI-FLO, Boston Scientific, Marlboro, MA, USA. Through the microcatheter, an emulsion of chemotherapeutics and iodized oil was slowly administered into the tumor-feeding arteries. Whenever it was physically possible, the ultra-selective TACE method was used.

Anti-cancer medications and embolic agents such as doxorubicin, (West-Ward Pharmaceuticals, Eatontown, NJ, USA), cisplatin (WG Critical Care LLC, Paramus, NJ, USA), Epirubicin (Pfizer, New York, NY, USA), mitoxantrone (Hospira Inc, Lake Forest, IL, USA), and mitomycin C (Accord Healthcare Inc, Durham, NC, USA) were combined and administered when the catheter was positioned in the branches of the artery that supply blood to the tumor; in most cases, the treatment included oxaliplatin (Sanofi; sanofi-aventis U.S. LLC, Bridgewater, NJ, USA) (50–100 mg) and pirarubicin (Sanofi) (10–40 mg) in combination with Lipiodol^R^ (Guerbet, Milan, Italy) (2–20 mL). The drug was chosen at the discretion of the GI oncologist on a case-by-case basis, and the dose of lipiodol was determined by the size of the tumor. If necessary, a gelatin sponge or polyvinyl alcohol particles, (300–500 μm) were injected to embolize the tumor-feeding arterioles until repeat angiography revealed no tumor staining. Gelatin sponge particles (Pfizer), were not employed in patients with tumor thrombosis in the major portal branch and/or Child-Pugh B liver function. Six to eight weeks after the surgery a contrast-agent-enhanced CT/MRI was performed.

To ensure that the entire tumor had been treated, further X-rays were performed. After that, the interventional radiologist removed the catheter and applied pressure to halt any bleeding after the surgery was finished. A closure device was used in some cases to reseal the hole in the artery. Depending on which artery was accessed and if a closure device was used, the patients were expected to spend two to six hours in the recovery room. Our patients’ operative time for the TACE procedures was about 90 min.

TACE was repeated on demand when remaining viable tumors were confirmed or new lesions appeared in patients with sufficient hepatic function.

All patients received comprehensive information about TKI agents, especially sorafenib, including its efficacy, side effects, and costs. Sorafenib was the only TKI agent used. Physicians advised patients to use sorafenib in addition to TACE; however, the final treatment decision was usually determined by the patient or their family members. Within 3–5 days after the first TACE sorafenib was given orally twice daily at a dose of 400 mg, then stopped the day before the next TACE procedure and restarted within 3–5 days following each TACE. When a grade 3 or 4 toxicity was detected, according to the National Cancer Institute Common Toxicity Criteria Adverse Events (CTC AE) version 3.0 (accessed on 12 December 2021) the dose of sorafenib was lowered or treatment was delayed or temporarily terminated to ensure optimal patient safety. The treatment was continued until either disease progression, unacceptable toxicity developed, or the patient deteriorated.

### 2.4. Tumor Size Criteria and Explant Pathology

All patients underwent an explant pathology review to determine the tumor size and percent of tumor necrosis at the time of their transplant. Patients with 100% tumor necrosis had no viable tumor upon pathology examination and were labeled as Milan criteria. For Milan criteria to be met tumor diameter of a single lesion must be less than or equal to 5 cm. For multiple lesions, Milan criteria allow for no more than 3 tumors each less than or equal to 3 cm. Both single and multiple lesions must not have a vascular invasion or extra-hepatic metastases [28]. The beyond Milan criteria are not as extreme as that in Milan but it depends on the overall tumor size, the number of nodules, different tumor markers such as AFP may utilize [28].

### 2.5. Unresectable HCC Patients

Unresectable HCCs are liver tumors that were not acceptable for surgical resection due to the extent, tumor burden, or anatomical location of the tumor in the liver. Additionally, they include patients over the age of 75, cirrhotic patients with low liver preservation, or those who refused surgical interventions.

### 2.6. Follow-Up

Patients had monthly follow-up blood profiles, including complete blood cell count and prothrombin time along with liver function tests of alanine aminotransferase (ALT), aspartate aminotransferase (AST), bilirubin, albumin, and serum alpha-fetoprotein (AFP). In addition, radiological scans included liver contrast-enhanced CT or MRI and, if necessary, a chest CT and/or bone scan was repeated every 6–9 weeks to evaluate the treatment response for HCC. The tumor response was assessed by using the modified Response Evaluation Criteria in Solid Tumors (mRECIST). The last follow-up visit included in this report occurred on 20 December 2019. The overall survival was measured from the date of starting the TACE treatment to the date of death from any cause or the last visit. Data from patients who lost to follow-up visits was utilized to include the last date they were known to be alive, Where the patients who have survived were censored at the data cutoff, which is 20 December 2019.

### 2.7. Statistical Analysis

Demographic and clinical data, as well as HCC recurrence post-transplant, were reported as frequencies and proportions for categorical variables and as the median and interquartile range (IQR) or mean (standard deviation (SD)) for continuous variables. The difference between groups was compared using the Chi-square or Fisher’s exact tests for categorical variables and the Kruskal-Wallis test or unpaired t-test for continuous variables. Percent necrosis by TACE + TKI versus TACE alone based on the pathologic group and the radiologic group was presented by the box plots. The difference in percent necrosis between groups was compared using the unpaired t-test. Overall patient survival and HCC-free survival were presented by the Kaplan-Meier curves. Differences between groups were compared using the log-rank test.

Generalized linear models (GLMs) were used to determine the characteristics associated with the percent of tumor necrosis on explant pathology. Cox proportional hazards modeling was used to determine the characteristics associated with overall mortality. Variables were selected based on potential clinical relevance via the Stata’s Lasso technique with the cross-validation (CV) selection option [29,30] and also based on their clinical importance. All analyses were performed on Stata version 17.0 (StataCorp LLC, College Station, TX, USA). A *p*-value of <0.05 was considered statistically significant.

## 3. Results

A total of 128 patients received an LT between July 2008–December 2019. Those 128 patients met the inclusion criteria, of which 95 were within the Milan criteria and 36 met the beyond Milan criteria (Figure 1). Patients had a median (IQR) age of 61.4 years (57.0, 66.3) and most were males (77%; Table 1). Patients who received TACE plus TKI, irrespective of tumor size, had higher biological Model for End-Stage Liver Disease scores compared to the TACE-alone group (18 [10,24]) vs. 12 [9,17], respectively, *p* = 0.02), but this difference was lost when accounting for HCC MELD exception points provided to the Milan criteria patients. The underlying disease etiology varied between the treatment groups in that most were hepatitis-C-seropositive for patients who received TACE, only 65% vs. 40% in the TACE plus TKI group, *p* = 0.02. In addition, more patients in the TACE-alone group had a diagnosis of diabetes mellitus compared to the TACE + TKI group, *p* = 0.02. A similar number of LRT procedures occurred between the treatment groups. Within the TACE-alone group, 79 (77%) patients met the Milan criteria and 24 (23%) met the beyond Milan criteria, while the TACE + TKI group had a higher proportion of patients in the beyond Milan size category, 16 (64%) vs. 9 (36%; *p* = 0.01).

### 3.1. Tumor Necrosis

The biggest clinical impact on tumor necrosis occurred in patients with lesions beyond Milan, in which a greater proportion of patients received a TKI agent (sorafenib) in combination with TACE. Therefore, there was the separation of treatment effects in patients with the beyond Milan criteria. Patients with lesions beyond the Milan criteria had an average amount of percent of tumor necrosis on explant pathology of 43.8% ± 32% for those who received TACE alone compared to 69.6% ± 32.8% for patients who received combination therapy of TACE + TKI, *p* = 0.03 (Figure 2). A statistically significant improvement in the percent of tumor necrosis was not seen based on the treatment groups in patients with Milan criteria tumors (TACE only 80% necrosis vs. TACE + TKI 95% necrosis, *p* = 0.22).

### 3.2. Univariate and Multivariate Analysis

A univariate analysis helped determine important characteristics associated with the percent of tumor necrosis to include in general linear modeling. Demographics, including age, gender, and race/ethnicity, body mass index at transplantation, history of diabetes, underlying disease etiology, MELD score at transplantation, radiographic tumor size, number of LRTs, AFP at listing or transplant, tumor size (cm) regression from listing to transplant, evidence of tumor progression, and largest pathologic focal size (cm) were not significantly associated with percent of tumor necrosis. On the other hand, the percent of tumor regression from listing to transplant, number of nodules, total pathologic tumor size, evidence of vascular invasion, and tumor size classification (Milan vs. beyond Milan) were significantly associated with tumor necrosis.

The multivariate model controls for the treatment group (TACE alone or TACE + TKI), recipient age, tumor size classification, and vascular invasion. The treatment group (*p* = 0.06) and recipient age (*p* = 0.08) did not significantly contribute to the model; however, having a beyond Milan tumor size classification and evidence of vascular invasion resulted in a reduction in tumor necrosis, −13.99 (95% CI −27.37, −0.60; *p* = 0.04) and −34.37 (−55.80, −12.95; *p* = 0.002), respectively (Table 2).

### 3.3. Beyond Milan Subanalysis: Disease-Free and Overall Survival

Because the benefit of combination therapy was more evident beyond Milan patients (*n* = 36) (Table 3), the remaining outcome analyses focused only on the beyond Milan group. The five-year disease-free survival (DFS) between the treatment groups approached significance, with 100% DFS in the TACE + TKI group vs. 67.2% in the TACE-alone group, *p* = 0.07 (Figure 3). No difference in five-year patient survival was seen between the treatment groups, 77.5% in the TACE + TKI group compared to 61.5% in the TACE-alone group (Figure 4). We further investigated the clinical characteristics associated with recurrence and death in patients beyond Milan tumors. None of the clinical characteristics used in the univariate analysis contributed to post-liver transplantation recurrence. Despite not seeing a difference in overall survival improvement, key characteristics varied between those who remained alive (*n* = 26) compared to those who died (*n* = 10), at least 5 years post-liver transplantation. Patients who died had a greater reduction in tumor size from listing to post-liver transplantation, −24.6% vs. 45.4%, *p* = 0.001. Vascular invasion occurred in 50% of deceased patients (*n* = 5), compared to only 8% of patients who remained alive (*n* = 2). No differences existed based on demographics, MELD, etiology, AFP at listing or transplant, radiographic or pathologic tumor size at listing or transplant, tumor growth, number of nodules, number of LRTs, largest tumor size on explant, total tumor size on explant, or waiting time from listing to transplant.

Vascular invasion was also explored. Seven patients had vascular invasion compared to twenty-nine who did not. No significant clinical characteristics were identified between patients with vascular invasion; however, it is worth mentioning that zero patients who received TACE + TKI had evidence of vascular invasion.

## 4. Discussion

In patients with early to intermediate-stage HCC, TACE has been demonstrated to improve survival. Most clinical practice guidelines consider it to be the standard of care and a recommended therapy modality. TACE procedures are frequently repeated due to large tumor volumes and high progression rates. Repeated TACE may result in the impairment of liver function, which is usually associated with the worst outcomes and a bad prognosis for patients. Gastrointestinal oncologists and interventional radiologists are both interested in the treatment of TACE-refractory HCC. Clinical trials combining TACE with other liver-directed and systemic therapies, such as ablation (e.g., radiotherapy, multi-kinase inhibitors, and immunotherapy, have been conducted to improve the efficacy of TACE. In one example TACE and sorafenib, a multi-kinase inhibitor was used concurrently as a trial treatment for unresectable HCC. However, the results were inconclusive. A recent study found that combining sorafenib with TACE significantly delays the recurrence of HCC. On the other hand, some trials did not initially show a significant benefit; however, later subgroup analyses suggested that a longer duration of treatment with sorafenib in combination with TACE may improve clinical outcomes.

The current retrospective study found that combining TACE and sorafenib significantly improved overall survival compared to TACE alone in patients with unresectable HCC. The advantage of combined TACE plus TKI therapy was more obvious beyond Milan patients (*n* = 36); therefore, the subsequent outcome analyses are limited to this group. The difference in five-year disease-free survival (DFS) between the treatment groups was statistically recognized, with 100% DFS in the TACE + TKI group vs. 67.2% in the TACE-alone group (*p* = 0.07). There was no difference in five-year patient survival between treatment groups, with 77.5% in the TACE + TKI group and 61.5% in the TACE-alone group.

Clinical variables linked to recurrence and death in patients with beyond Milan tumors were also studied. None of the clinical factors utilized in the univariate analysis were associated with LT recurrence after liver transplantation. Even though there was no difference in overall survival, crucial characteristics differed between those who remained alive (*n* = 26) and those who died. At least 5 years post-liver transplantation (*n* = 10), patients who died had a higher drop in tumor size from listing to post-liver transplantation: −24.6 percent vs. 45.4 percent, *p* = 0.001. Having a beyond Milan tumor size categorization and evidence of vascular invasion resulted in a reduction in tumor necrosis of −13.99 (95% CI −27.37, −0.60; *p* = 0.04) and −34.37 (−55.80, −12.95), *p* = 0.002, respectively. The greatest clinical impact on tumor necrosis occurred in patients with lesions beyond Milan, where a higher proportion of patients received sorafenib in combination with TACE. In patients with Milan criteria tumors, there was no statistically significant improvement in percent tumor necrosis based on treatment groups (80 percent necrosis vs. 95 percent, *p* = 0.22). Beyond the Milan criteria, there is a separation of the impact of therapy on patients. Patients with tumors that met the beyond Milan criteria showed an average percent of tumor necrosis on explant histology of 43.8 percent 32 percent for those who received TACE alone versus 69.6 percent for 32.8 percent for those who received TACE + TKI, *p* = 0.03. Regardless of tumor size, patients who received TACE + TKI had higher biological Model for End-Stage Liver Disease scores than those who received TACE alone (18 [10,24] vs. 12 [9,17], *p* = 0.02), but this advantage was lost when HCC MELD exception points were given to Milan criteria patients. The most underlying disease etiology among the patient sample was hepatitis-C-seropositive, 65 percent of TACE only vs. 40 percent in the TACE plus TKI group, *p* = 0.02. Furthermore, the TACE-alone group had more patients with diabetes mellitus than the TACE + TKI group, *p* = 0.02. In both therapy groups, the number of LRT procedures was similar. The TACE-alone group had 79 (77%) patients who fulfilled the Milan criteria and 24 (23%) patients who met the beyond Milan criteria, but the TACE + TKI group had a greater proportion of patients in the beyond Milan size category, 16 (64%) vs. 9 (36%; *p* = 0.01).

Whereas the current report is a retrospective study on a very good number of patients, it suggests and highlights that using TACE plus TKI in unresectable HCC patients is associated with improved overall survival in cancer that is known for a short recurrence. We believe that offering TACE plus TKI to HCC patients beyond the Milan criteria might provide better outcomes and bridge more patients to curative therapy of liver transplantation.

A cohort study with a prospective follow-up and a protocol timeline for successive blood collection and other investigational tools such as scans and ctDNA may be necessary for the future to demonstrate the efficacy of using TCAE plus TKI in HCC patients who undergo LT. A prospective clinical trial at our institution (Houston Methodist Cancer Center and JC Walter Jr Center for Transplantation) is currently being conducted to address the above clinical need (NCT05171335) [31].

## 5. Conclusions

The current study found that using TACE plus sorafenib is generally well-tolerated and associated with improved overall survival in transplant recipient patients with unresectable HCC. A multi-center and prospective study is needed. Randomized and controlled trials are needed to confirm these preliminary findings.

## Figures and Tables

**Figure 1 cancers-14-00650-f001:**
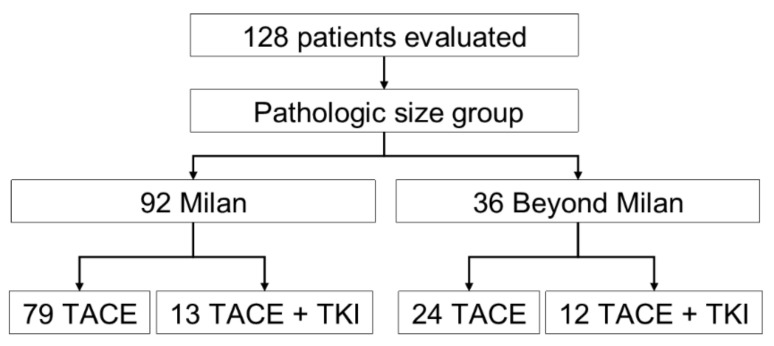
Flowchart of the study population, TACE, transarterial chemoembolization; TKI, tyrosine kinase inhibitor.

**Figure 2 cancers-14-00650-f002:**
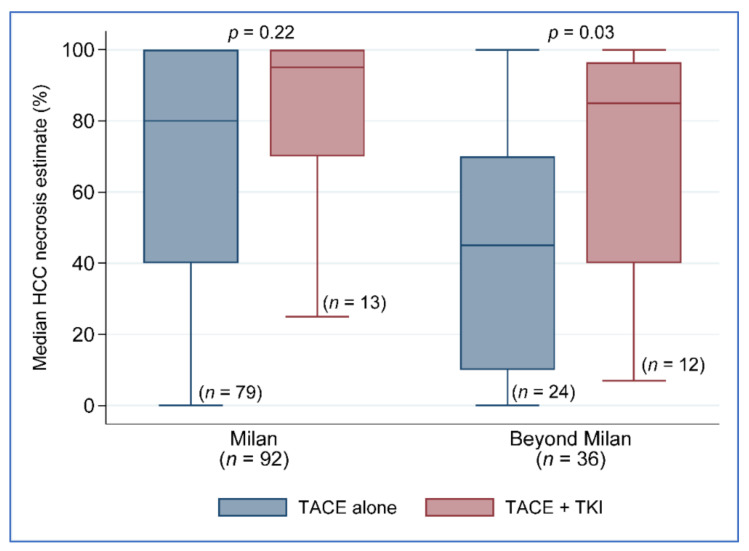
Percent necrosis by treatment group and tumor size classification. TACE, transarterial chemoembolization; TKI.

**Figure 3 cancers-14-00650-f003:**
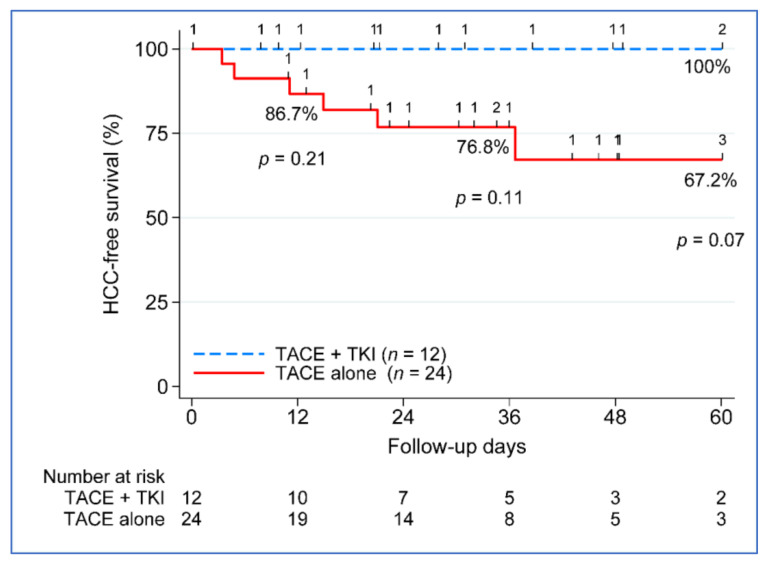
Five-year disease-free survival post-liver transplant. TACE, transarterial chemoembolization; TKI, tyrosine kinase inhibitor; and *n*, number.

**Figure 4 cancers-14-00650-f004:**
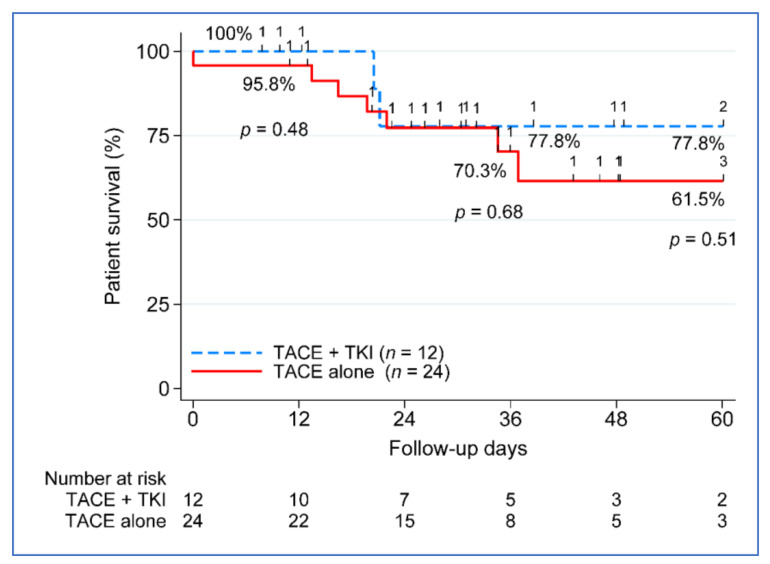
Five-year patient survival post-transplant. TACE, transarterial chemoembolization; TKI, tyrosine kinase inhibitor; and *n*, number.

**Table 1 cancers-14-00650-t001:** Patient demographics were stratified by the TACE group.

Characteristics	Total	TACE Alone	TACE + TKI	*p*-Value
	(*n* = 128)	(*n* = 103)	(*n* = 25)	
Age (year), median (IQR)	61.4 (57.0, 66.3)	61.5 (57.2, 66.3)	61.0 (54.6, 66.1)	0.58
Male gender	98 (76.6)	78 (75.7)	20 (80.0)	0.65
Race/ethnicity				0.03
Caucasian	82 (64.1)	71 (68.9)	11 (44.0)	
African American	11 (8.6)	9 (8.7)	2 (8.0)	
Hispanic	28 (21.9)	17 (16.5)	11 (44.0)	
Asian	7 (5.5)	6 (5.8)	1 (4.0)	
BMI at transplant, median (IQR)	27.2 (24.4, 30.7)	26.8 (24.4, 30.6)	27.5 (24.2, 31.1)	0.62
History of diabetes mellitus	56 (43.8)	40 (38.8)	16 (64.0)	0.02
Underlying disease etiology *				
Hepatitis-C-seropositive	77 (60.2)	67 (65.0)	10 (40.0)	0.02
Hepatitis B virus	5 (3.9)	3 (2.9)	2 (8.0)	0.24
ETOH	16 (12.5)	13 (12.6)	3 (12.0)	0.93
NASH/Crypto	20 (15.6)	15 (14.6)	5 (20.0)	0.50
Others	14 (10.9)	9 (8.7)	5 (20.0)	0.11
Biological MELD at transplant, median (IQR)	13 (9, 19)	12 (9, 17)	18 (10, 24)	0.02
Exception MELD at transplant, median (IQR)	29 (27.5, 32)	29 (27, 31)	31 (28, 33)	0.22
Waiting time from listing to transplant (days), median (IQR)	343.0 (190.0, 499.0)	339.0 (190.0, 510.0)	385.0 (157.0, 479.0)	0.90
Tumor classification (pathological)				0.01
Inside the Milan criteria	92 (71.9)	79 (76.7)	13 (52.0)	
Beyond Milan	36 (28.1)	24 (23.3)	12 (48.0)	
Total number of LRT				0.63
1	74 (57.8)	62 (60.2)	12 (48.0)	
2	41 (32.0)	31 (30.1)	10 (40.0)	
3	10 (7.8)	7 (6.8)	3 (12.0)	
4	2 (1.6)	2 (1.9)	0 (0.0)	
5	1 (0.8)	1 (1.0)	0 (0.0)	

* Patient may have multiple underlying etiologies. Values are in number and % unless otherwise indicated. IQR, interquartile range; BMI, body mass index; MELD, Model for End-Stage Liver Disease; LRT, locoregional therapy; TACE, transarterial chemoembolization; TKI, tyrosine kinase inhibitor; RFA, radiofrequency ablation; ETOH, alcoholic; and NASH, nonalcoholic steatohepatitis.

**Table 2 cancers-14-00650-t002:** Characteristics associated with the percent of tumor necrosis on explant pathology.

Characteristics	Multivariable GLM	
	(95% CI)	*p*-Value
Tumor control therapy		
TACE alone	(Reference)	
TACE + TKI	14.26 (−0.63, 29.15)	0.06
Age (year)	0.75 (−0.07, 1.57)	0.08
Tumor classification (pathological)		
Inside the Milan criteria	(Reference)	
Beyond Milan	−13.99 (−27.37, −0.60)	0.04
Vascular invasive	−34.37 (−55.80, −12.95)	0.002

TACE, transarterial chemoembolization; TKI, tyrosine kinase inhibitor; GLM, generalized linear modeling and CI, confidence interval.

**Table 3 cancers-14-00650-t003:** Patient demographics of the beyond Milan group only, stratified by TACE group.

Characteristics	Total	TACE Alone	TACE + TKI	*p*-Value
	(*n* = 36)	(*n* = 24)	(*n* = 12)	
Age (year), median (IQR)	60.9 (55.4, 66.1)	60.9 (56.4, 65.9)	62.3 (54.4, 66.3)	0.76
Male gender	32 (88.9)	22 (91.7)	10 (83.3)	0.45
Race/ethnicity				0.01
Caucasian	26 (72.2)	21 (87.5)	5 (41.7)	
African American	1 (2.8)	0 (0.0)	1 (8.3)	
Hispanic	9 (25.0)	3 (12.5)	6 (50.0)	
Asian	0 (0.0)	0 (0.0)	0 (0.0)	
BMI at transplant, median (IQR)	27.7 (25.3, 30.5)	27.7 (25.3, 30.5)	27.9 (25.1, 30.5)	0.83
History of diabetes mellitus	20 (55.6)	12 (50.0)	8 (66.7)	0.34
Underlying disease etiology *				
Hepatitis-C-seropositive	20 (55.6)	15 (62.5)	5 (41.7)	0.24
Hepatitis B virus	1 (2.8)	0 (0.0)	1 (8.3)	0.15
ETOH	6 (16.7)	5 (20.8)	1 (8.3)	0.34
NASH/Crypto	2 (5.6)	1 (4.2)	1 (8.3)	0.61
Others	7 (19.4)	3 (12.5)	4 (33.3)	0.14
HBcAb	9 (25.7)	5 (21.7)	4 (33.3)	0.46
Biological MELD at transplant, median (IQR)	15 (9.5, 25)	12.5 (9, 22)	22 (12, 31)	0.12
Exception MELD at transplant, median (IQR)	29.5 (27, 33)	29 (27, 33)	31.5 (27, 35)	0.40
Waiting time from listing to transplant (days), median (IQR)	270.5 (140.0, 469.0)	319.0 (139.5, 440.5)	197.0 (140.0, 504.5)	0.76
Total number of LRT				0.61
1	12 (33.3)	8 (33.3)	4 (33.3)	
2	18 (50.0)	13 (54.2)	5 (41.7)	
3	6 (16.7)	3 (12.5)	3 (25.0)	

* Patient may have multiple underlying etiologies. Values are in number and % unless otherwise indicated. IQR, interquartile range; BMI, body mass index; AFP, alpha-fetoprotein; MELD, Model for End-Stage Liver Disease; LRT, locoregional therapy; TACE, transarterial chemoembolization; TKI, tyrosine kinase inhibitor; RFA, radiofrequency ablation; ETOH, alcoholic, NASH, nonalcoholic steatohepatitis and HBcAb, hepatitis B core antibody.

## Data Availability

The data of this study that support our results are available on request from the corresponding author, Maen Abdelrahim.

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
