# Peer review of "Transarterial Chemoembolization (TACE) Plus Sorafenib Compared to TACE Alone in Transplant Recipients with Hepatocellular Carcinoma: An Institution Experience"

_cancers, 2022, doi:10.3390/cancers14030650_

Round 1

Reviewer 1 Report

The authors revealed the possibility that TACE plus TKI combination therapy for the patients with unresectable HCC awaiting LT improved overall survival compared to TACE monotherapy. This study is interesting and very important for the patients with HCC awaiting LT. However, there are some issues for acceptance.

  1. This study demonstrated 100 % DFS at 5-year in the TACE plus TKI group; however, 77.8 % survival at 5-year in the same group. Why the patients in this group died? Why is there no survival benefit in the TACE plus TKI group, although this group demonstrated 100 % DFS at 5-year?

  1. In this study, DFS and survival on the Beyond Milan group were analyzed, thus the authors should present patient demographics as Table 1.

  1. From this study findings, is the TACE plus TKI combination therapy recommended to only patients with Beyond Milan? The author should discuss this point in Discussion.

Are DFS and OS different in the patients with Milan who received TACE plus TKI combination therapy or TACE monotherapy?

  1. In Materials and Methods, the authors should add the description “unresectable HCC”.

Author Response

Dear reviewer,

Thank you for all your comments and  I would like to provide a short cover letter detailing our changes attached furthermore, we have attached the updated manuscript including your comments.

Sincerely,

C. author   

Reviewer 2 Report

The manuscript is interesting; the efficacy of TACE+sorafenib in HCC patients is still matter of debate but little is known in transplanted patients. 

I have the following comments:

1) How was the treatment strategy decided? Why a group of patients was treated with the combined therapy and another group with TACE alone?

2) More technical details concerning TACE procedures should be provided. 

3) A brief comment on the controversial effect of the chemoterapeutic agent injected during TACE should be added (cite the meta-analysis PMID: 28588882)

4) Please add censored data to the Kaplan Meyer curves

Author Response

(The authors gave the same response as above.)

Round 2

Reviewer 1 Report

Authors correctly responded to the reviewer’s comments.

Reviewer 2 Report

The manuscript is OK in the current form.